# Legume NCRs and nodule-specific defensins of actinorhizal plants—Do they share a common origin?

**Marco Guedes Salgado**[1], **Irina V. Demina**[1], **Pooja Jha Maity**[1], **Anurupa Nagchowdhury**[1], **Andrea Caputo**[1¤a], **Elizaveta Krol**[2,3], **Christoph Loderer**[4¤b], **Günther Muth**[5], **Anke Becker**[2,3], **Katharina Pawlowski**[1]*

1 Department of Ecology, Environment and Plant Sciences, Stockholm University, Stockholm, Sweden, 2 Center for Synthetic Microbiology, Philipps-Universität Marburg, Marburg, Germany, 3 Department of Biology, Philipps-Universität Marburg, Marburg, Germany, 4 Department of Biochemistry and Biophysics, Stockholm University, Stockholm, Sweden, 5 Department of Microbial Bioactive Compounds, Interfaculty Institute of Microbiology and Infection Medicine (IMIT), Eberhard Karls University Tübingen, Tübingen, Germany

¤a Current address: Department of Medical Sciences, Uppsala University, Uppsala, Sweden
¤b Current address: Institute for Microbiology, Technische Universität Dresden, Dresden, Germany
* katharina.pawlowski@su.se

**Data Availability Statement:** NCBI accession number DgDef1 promoter: MZ779183 DgDef gene family: HQ005271.2 JX912726 MN388819 MN388820 MN388821 MN388822 Differential

## Abstract

The actinorhizal plant *Datisca glomerata* (Datiscaceae, Cucurbitales) establishes a root nodule symbiosis with actinobacteria from the earliest branching symbiotic *Frankia* clade. A subfamily of a gene family encoding nodule-specific defensin-like cysteine-rich peptides is highly expressed in *D. glomerata* nodules. Phylogenetic analysis of the defensin domain showed that these defensin-like peptides share a common evolutionary origin with nodule-specific defensins from actinorhizal Fagales and with nodule-specific cysteine-rich peptides (NCRs) from legumes. In this study, the family member with the highest expression levels, *DgDef1*, was characterized. Promoter-*GUS* studies on transgenic hairy roots showed expression in the early stage of differentiation of infected cells, and transient expression in the nodule apex. DgDef1 contains an N-terminal signal peptide and a C-terminal acidic domain which are likely involved in subcellular targeting and do not affect peptide activity. *In vitro* studies with *E. coli* and *Sinorhizobium meliloti* 1021 showed that the defensin domain of DgDef1 has a cytotoxic effect, leading to membrane disruption with 50% lethality for *S. meliloti* 1021 at 20.8 µM. Analysis of the *S. meliloti* 1021 transcriptome showed that, at sub-lethal concentrations, DgDef1 induced the expression of terminal quinol oxidases, which are associated with the oxidative stress response and are also expressed during symbiosis. Overall, the changes induced by DgDef1 are reminiscent of those of some legume NCRs, suggesting that nodule-specific defensin-like peptides were part of the original root nodule toolkit and were subsequently lost in most symbiotic legumes, while being maintained in the actinorhizal lineages.

expression data for Sinorhizobium meliloti 1021: https://www.ebi.ac.uk/arrayexpress/ (accession number E-MTAB-11181).

**Funding:** KP, VR 2012-03061, Vetenskapsrådet, vr. se AB, access to the resources of BMBF grant FKZ 031A533 (network grant for de.NBI); www.bmbf.de The funders had no role in study design, data collection and analysis, decision to publish, or preparation of the manuscript.

**Competing interests:** The authors have declared that no competing interests exist.

## Introduction

The production of antimicrobial peptides by plants is part of the defence against pathogens, playing a key role in innate immunity. These peptides' mode of action typically involves disruption of the plasma membrane of the pathogen [1]. Antimicrobial peptides include several classes of cysteine-rich peptides which are characterized by the number and spacing of their cysteine residues, and the disulfide bridges formed by them [2,3]. One of these classes are the defensins that act against viruses, bacteria and fungi [4,5]. They represent a group of small (<100 amino acids), cationic, highly stable cysteine-rich antimicrobial peptides (AMPs) organized in three antiparallel beta-strands and one alpha-helix, stabilized by four disulfide bridges [5]. Defensins can be produced through the course of development, where they can be either involved in regulation of plant growth and cellular signaling [6,7] or in plant responses to biotic and abiotic stresses [8]. For their antimicrobial activity, defensins may enter microbes through transient pores and initiate molecular responses by specific targeting [9]. Defensins have been divided into two classes: the precursors of members of the largest class (class I) contain signal peptides that target the active form to the extracellular space; members of class II exhibit an additional C-terminal domain, with variable length, which is involved in either vacuolar targeting or in host protection against the toxicity of the mature peptide [10]. Cysteine-rich peptides in general, and defensins in particular, have been identified in all plant organs examined, amongst others in nitrogen-fixing root nodules.

Nitrogen is the element that most often limits plant growth, and only some prokaryotes can form the enzyme complex nitrogenase to reduce air dinitrogen to ammonia for introduction into the biosphere. Two types of intracellular symbioses between higher plants and nitrogen-fixing soil bacteria are known: legume/rhizobia symbioses and actinorhizal symbioses. The latter are entered between a diverse group of plants, collectively called actinorhizal plants, and nitrogen-fixing Gram-positive soil actinobacteria of the genus *Frankia* [11]. Both types of intracellular symbioses go back to a common origin, the ancestor of the Fabales, Fagales, Cucurbitales and Rosales, however the symbiotic trait was lost in the majority of the lineages descended from this ancestor [12,13]. The reasons behind the counter-selection of root nodule symbioses are still under debate [13,14]. Indeed, since the symbiosis represents a valuable source of nitrogen to the plant, and the internal accommodation of the microsymbionts is largely controlled by the host plant itself, what are then the factors that have played a role towards counter-selection of root nodule symbiosis? One hypothesis is that "cheating" microsymbionts–*i.e.*, microsymbionts that require more carbon input then the fixed nitrogen they provide is worth–were a factor making a symbiosis unfavourable [13,15,16]. Another potential disadvantage of a root nodule symbiosis is the effect of nodule formation on plant immunity [17]. Nodules offer the microsymbionts a protected niche for propagation. This massive bacterial colonization is not associated with plant defence responses, indicating a down-regulation of defence mechanisms. However, the suppression of plant immunity in nodules to allow massive rhizobial colonization, would mean an increased vulnerability of nodules to phytopathogenic and opportunistic microbes [18].

Therefore, it is not surprising that many types of cysteine-rich peptides have been identified in root nodules. Legumes produce nodule-specific cysteine-rich peptides (NCRs) which are lethal to a variety of Gram-negative and Gram-positive strains in culture, disrupting the integrity of their plasma membranes [19,20]. NCRs also affect the differentiation of the intracellular rhizobia in nodules by inducing endoreduplication, thereby rendering them unable to survive outside nodules, and affect the permeability of their plasma membranes in a way to promote the exchange of nitrogenous solutes [21–23]. These peptides occur in nodules of members of the Inverted Repeat-Lacking Clade (IRLC) of legumes, such as *Medicago truncatula* or *Pisum*

*sativum*, whose genomes contain hundreds of different *NCR* genes [24,25]. They are also found in a lineage of legumes belonging to the Dalbergoid clade, namely in *Aeschynomene* spp., which has been ascribed to convergent evolution [26].

No orthologues of NCRs have been identified in actinorhizal nodules, but in two species of actinorhizal Fagales, *Alnus glutinosa* (Betulaceae) and *Casuarina glauca* (Casuarinaceae), small families of nodule-specific defensins have been identified; one representative from *A. glutinosa*, Ag5, has been characterized and was shown to affect the integrity of microsymbiont membranes, leading to leakage of nitrogenous solutes, an effect similar to that achieved by legume NCRs [27]. Transcripts of nodule-specific defensins were also identified in *Datisca glomerata* (Datiscaceae, Cucurbitales) and *Ceanothus thyrsiflorus* (Rhamnaceae, Rosales) [28,29].

To understand whether legume NCRs and nodule-specific defensins from actinorhizal nodules represent a case of convergent evolution or shared origin, we analysed the phylogeny, expression pattern and function of the previously reported nodule-specific defensin DgDef1 from the actinorhizal plant *D. glomerata* (Cucurbitaceae, Cucurbitales) [28]. *D. glomerata* is a suffruticose plant native to California and Northern Mexico that is nodulated by members of a *Frankia* clade (cluster-2) which mostly consists of uncultured strains [30–32]. Therefore, the effect of DgDef1 was investigated using another soil actinobacterium, *Streptomyces coelicolor* A3(2) M145, and two Gram-negative strains, *E. coli* K-12 substrain MG1655 and the legume microsymbiont *Sinorhizobium meliloti* 1021.

## Material and methods

### Biological material and growth conditions

*Datisca glomerata* (C. Presl.) Baill. seeds were collected from greenhouse plants originating from plants growing in Vaca Hills (California, USA). Seeds were germinated on sand wetted with water; eventually, plantlets were transferred to pot soil (S-jord, Hasselfors Garden AB, Hasselfors, Sweden) and cultivated under a 13h photoperiod and day/night temperatures of 23°C/19°C, with a light intensity of 60–100 $\mu Em^{-2}s^{-1}$. When the plants had reached a height of *ca*. 20 cm, they were transferred to larger pots containing a mixture of 1:1 soil/sand (Rådasand, Lidköping, Sweden; 0.4–0.6mm) and were inoculated with a suspension of nodules (*ca*. 1 g nodules/L soil). The suspension was prepared from nodules of older *D. glomerata* plants ground in deioinized water with mortar and pestle ("crushed nodules"). Plants were watered twice a week, once with deionized water and once with ¼ strength Hoagland's medium [33] without a nitrogen source. *Nicotiana benthamiana* plants were germinated and grown on soil for agroinfiltration assays under the same growth conditions.

One shot TOP10 chemically competent *E. coli* cells (ThermoFisher; Göteborg, Sweden) were used for transformation. Selection took place on Luria-Bertani (LB) [34] plates with 100 µg/L ampicillin. *Pichia pastoris* SMD1168 (*his4*, *pep4*) was used for heterologous expression. *P. pastoris* media recipes were prepared as described in the *Pichia* Expression Kit (ThermoFisher, cat. no. K1710-01). *Agrobacterium tumefaciens* LBA4404 and *Agrobacterium rhizogenes* LBA1334 were grown on yeast extract beef (YEB) medium [35] with 50 µg/ml rifampicin. *Sinorhizobium meliloti* 1021 was grown in tryptone yeast extract (TY) medium [36] with 600 µg/L streptomycin.

### Phylogeny of Cys-rich domains of defensins and legume nodule-specific cysteine-rich peptides (NCRs)

To reconstruct the phylogeny of the putative active domain of DgDef1 (GenBank: AEK82126.2), two defensins from pea (GenBank: ACI15746.1; UniProtKB: Q01784.1), two

defensins from soybean (NCBI Reference Sequence: XP_006586320.1; GenBank: KAG5014386.1), and a defensin from chickpea (GenBank: AAO38756.1) were used to generate a HMMER profile, which was then used to query the UniProt reference proteomes database (E-value = 10e-19), retrieving a total of 165 peptides. To this set, six peptides from *Datisca glomerata* [28,29], one peptide from *Ceanothus thyrsiflorus* [29], nine peptides from members of the Fagales [27,37] as well as six peptides from the genus *Aeschynomene* [26] were added. After manual curation, 80 peptides remained for phylogenetic reconstruction. Sequences were aligned with ProbCons v1.12 [38] and well-aligned positions were selected with BMGE using the BLOSUM62 substitution matrix [39]. The phylogenetic tree was estimated using RAxML v.8.2.12 [40] using the "PROTGAMMAAUTO" model and rapid bootstrapping where bootstrap replicates were automatically stopped upon convergence with autoMRE bootstopping [41].

## Gene expression analysis

Differential gene expression was assessed by Real Time quantitative PCR (RT-qPCR) as described in Salgado et al. [29]. Transcript abundance of genes coding for members of the nodule-specific subfamily of defensin-like peptides was assessed in roots and nodules of *D. glomerata*.

For validation of RNAseq results on *S. meliloti* cultures, RT-qPCR measurements were performed in a qTOWER3 G (Analytik Jena) using Power SYBR® Green RNA-to-CT™ 1-Step Kit (Thermo Scientific) according to the user manual in 16 µl reaction volume with 50 ng total RNA as template and 500 nM primers, in three biological and three technical replicates. Statistics were calculated in RStudio [42]. All primers used are listed in S1 Table.

## Construction of *GFP* fusions of the signal peptide and C-terminal domain of *DgDef1* for *Agrobacterium tumefaciens*-mediated transient transformation of *Nicotiana benthamiana*

Reporter constructs were generated by splice overlap PCR to create fused versions of *GFP*. The signal peptide of *DgDef1* was fused to the N-terminus of *GFP*, and the *CTTP* was fused to the C-terminus of *GFP*. *GFP* without added domains was used as a control. A scheme illustrating the different constructs is provided in S1B Fig. Primers used are listed in S1 Table. The *GFP* coding sequence was amplified based on H2-Venus (Addgene plasmid # 20971 [43]). Amplicons were first ligated into the restriction sites *Xho*I/*Bam*HI of the destination vector pUC18-entry8 [44], followed by insertion downstream of the 2x35S promoter in the binary vector pMDC132 *via* Gateway (ThermoFisher). *Agrobacterium tumefaciens* LBA4404 was transformed and agroinfiltration of *Nicotiana benthamiana* was performed following the method described by Pike et al. [45].

## Confocal microscopy

For imaging of plasmolyzed cells, epidermal peels of tobacco leaves were pre-treated with 750 mM Sorbitol, 10 mM MES for 45 min. All the preparations were treated with 0.005% of calcofluor-white, 2 min. Leaf peels were mounted onto a glass slide using one drop of ProLong® Diamond Antifade Mountant (Molecular Probes) and covered with a 170 µm thick coverslip (#1.5; VWR). Imaging took place in a Zeiss LM 800 confocal microscope equipped with the laser lines 405 (for the calcofluor signal) and 488 (for the EGFP signal). Preliminary bleed-through controls included i) simultaneous excitations at 405 and 488 nm on single dyed

samples and ii) confirmation of absence of fluorescence in non-target fluorophores (*e.g.*, 488 laser *vs*. Calcofluor-white, and vice versa). Micrographs were processed in IMARIS v.9.2 (Bitplane).

## Amplification of the *DgDef1* promoter

The promoter region of the gene *DgDEF1* was amplified from adaptor-ligated genomic libraries by genome walking using the GenomeWalker[TM] Universal Kit (TakaraBio, Mountain View, CA, USA). Genomic DNA from *D. glomerata* leaves was isolated according to Ribeiro et al. [46]. Per library, 2.5 μg of the DNA were digested, purified and ligated to GenomeWalkerTM Adaptors as described by the manufacturer (TakaraBio). Restriction enzymes *Eco*RV, *Sca*I, *Dra*I, *Pvu*II and *Stu*I were used for preparation of the genomic libraries DL1, DL2, DL3, DL4 and DL5. Gene-specific primers used for primary and secondary genome walking PCRs are listed in S1 Table. PCR amplification was conducted according to the GenomeWalker[TM] protocol (TakaraBio), except for one modification: the denaturation step was carried out at 94˚C for 15 s. The promoter region of *DgDEF1* was PCR amplified from DL1 with the primers proDgDEF1-for and proDgDEF1-rev (S1 Table). All PCR products were cloned in pJET1.2 (ThermoFisher) and subsequently sequenced (Eurofins) using the primers pJET1.2-for and pJET1.2-rev ThermoFisher) or gene-specific primers designed for sequencing (S1 Table).

## Preparation of promoter:*GUS* fusion construct

Using the Gateway cloning technology (TakaraBio), the promoter regions were transferred from the entry vector pUC18-entry8 [44] into the destination vector–an integration vector derived from pIV10 [47]–upstream of the reporter gene ORF to yield promoter:*GUS* fusions.

In order to clone the promoter fragment in the entry vector, forward and reverse primers (S1 Table) were designed to introduce the respective restrictions sites at the flanks of the promoter fragments. PCR was conducted on genomic DNA as described above. The *Bam*HI/*Not*I *DgDEF1* promoter fragment was subcloned in *Bam*HI/*Not*I-digested pUC18-entry8. The pGWB203 vector with a promoter:*GUS* construct was transferred into *A. rhizogenes* LBA1334 by electroporation and transformants were selected on 50 μg/ml kanamycin. The pIV10 vector was integrated into *A. rhizogenes* AR1193 TL-DNA segment in the course of triparental mating [47]. Selection of integration events was carried out on YEB agar medium containing 100 μg/ml ampicillin, 100 μg/ml spectinomycin and 100 μg/ml rifampicin. Selected transformants were confirmed by colony PCR or liquid culture PCR with a forward gene-specific primer and either the EcGUS-rev primer or the M13-rev primer (S1 Table).

## *Agrobacterium rhizogenes*-mediated transformation of *Datisca glomerata*

Experiments were repeated in four independent series. The first transformation was performed according to Markmann et al. [48] with some modifications. 10-Week-old plants were incubated for two days in the dark at 4˚C prior to inoculation with *A. rhizogenes* carrying a promoter:*GUS* construct. The inoculum (a paste of *A. rhizogenes* grown on YEB agar with selective antibiotics for 24h) was applied over needle-stabbed hypocotyls and plants were kept i) 2 days in the dark at 18˚C; ii) 4 days at 15h light/9h dark photoperiod, 18˚C; iii) at 15h light/9h dark photoperiod at 23˚C and 19˚C, respectively. When the hairy roots formed at the wound sites had reached a size that could support the shoot, the wild-type root was excised and the plants were transplanted to larger pots for inoculation with "crushed nodules". Inoculation was repeated after two weeks. During the period starting from *A. rhizogenes*-mediated transformation until inoculation with *Frankia*, the plants were watered with ¼ strength Hoagland's medium with nitrogen [33]. Upon inoculation, the plants were watered with ¼ strength

Hoagland's medium without nitrogen source. To prevent symptoms of nitrogen deprivation, occasionally the medium was supplemented with 1 mM KNO₃. The following transformations were performed according to Demina et al. [49].

To confirm the transformation of *D. glomerata* and evaluate for the viability of *A. rhizogenes* post-inoculation, genomic DNA was isolated from roots according to Edwards at al. [50]. PCR was then conducted to check i) the DNA integrity based on ubiquitin gene (*Dgc205*; [28]); ii) the genomic integration of the promoter:*GUS*; iii) the transfer of Ri plasmid T-DNA and *A. rhizogenes* survival by amplification of *rolB and virD*, respectively, as described elsewhere [51].

## Histochemical staining for β-glucuronidase (GUS) activity and documentation

Roots and nodules of *D. glomerata* were harvested and washed in GUS reaction buffer: either ¼ strength SB buffer (12.5 mM PIPES, 1.25 mM MgSO₄, 1.25 mM EGTA, pH 6.9) or 100 mM sodium phosphate buffer (pH 6.8) containing 1 mM EDTA, 0.1% (v/v) Triton X-100 with 0.25 mM K₃[Fe(CN)₆] or without added ferricyanide. Then, the samples were transferred to GUS reaction buffer containing 1 mM X-Gluc (5-bromo-4-chloro-3-indolyl-beta-D-glucuronide), vacuum-infiltrated three times, each time for 5 min and incubated for several hours up to over-night at 37°C in the dark. Afterwards, the samples were placed in ¼ strength SB or phosphate buffer containing 3% (w/v) paraformaldehyde, 0.1% (v/v) Tween 20 and 0.1% (v/v) Triton X-100, vacuum-infiltrated as described above and incubated overnight at 4°C. After fixation the samples were washed several times in reaction buffer. The fixed nodules were embedded in 3% (v/v) agarose and sectioned on a Leica VT1000E vibratome (Leica Biosystems, Wetzlar, Germany). Sections of 50–70 μm thickness were observed under an Axiovert 200M (Zeiss, Jena, Germany) using bright field microscopy; photographs were taken using an Axio HRC Camera (Zeiss).

## Preparation of an expression cassette for heterologous production of DgDef1 using the model yeast *Pichia pastoris*

The expression vector pPIC9K for secretion in *P. pastoris* was purchased from ThermoFisher (San Diego, CA). *In silico* analysis were carried out at the SignalP 4.1 server (http://www.cbs.dtu.dk/services/SignalP/) predicting an N-terminal signal peptide in the DgDef1 ORF. To prepare a synthetic expression cassette, a sense primer carrying a 5'-*EcoR*I-6His-Thrombin-3' and an antisense primer with a *Not*I restriction site were designed in order to amplify a truncated *DgDef1* ORF (*DgDef1ΔSP*), *i.e.*, compared with its full ORF, *DgDef1ΔSP* lacks the codons of its native N-terminal signal peptide sequence. The double-restricted 336 bp amplicon was inserted in frame downstream to the *Saccharomyces cerevisiae* α-mating factor secretion signal sequence (α-MF), between the *EcoR*I and *Not*I cloning sites of the pPIC9K vector (Thermo-Fisher). The correctness of the resulting plasmid pPIC9K-α-MF-6His-Thrombin-DgDef1ΔSP was confirmed by sequencing. The procedure is illustrated in S1A Fig.

## Selection for *Pichia pastoris* multicopy transformants

Prior to yeast transformation, 20 μg of plasmid pPIC9K-α-MF-6His-Thrombin-ΔDef1 were propagated and prepared after the Qiagen Midi prep kit. Analogously, an equal amount of an empty pPIC9K vector was prepared along to be used as a negative control in downstream expression studies. Before recombinant integration, both constructs were restricted with *Sac*I. The linearized product was integrated into the genome of Sorbitol-pretreated *P. pastoris* using an ECM600 (BTX) gene pulser instrument according to the conditions described by Becker et.

al. [52]. Electrotransformed yeast cells were spread onto MD plates and selected according to their ability to grow on Histidine-deprived media (His$^+$ phenotype). Subsequently, His$^+$ transformants were selected for multiple transgene insertions by a drug test: positive integrants were submitted to increasing amounts of antibiotic G418, ranging from 0.1–2.0 mg·ml$^{-1}$ in YPD-agar plates. Clones able to grow in G418 (1 mg·ml$^{-1}$) were evaluated by direct colony-PCR to screen its genomic integration. Only transformants with Methanol Utilization Plus phenotype (Mut$^+$) were chosen for small-scale time-course expression trials.

## Heterologous expression of DgDef1*ΔSP* in *Pichia pastoris*

During small-scale expression, selected clones grew in Buffered Glycerol complex Medium (BMGY; *Pichia* Expression Kit from ThermoFisher; pH 6.0) until an OD$_{600}$ between 2.0 and 6.0 had been reached. Cells were pelleted at 350 x *g*, 10 min, 15–25˚C and resuspended in 50 ml of BMMY induction medium (*Pichia* Expression Kit) supplied with methanol 0.5% (v/v) to a final density of OD$_{600}$ = 1.0. The culture was maintained at 28˚C, 220 rpm for 72 h. Methanol was added every 24 h to a final concentration of 0.5% (v/v). Expression was monitored by SDS-PAGE and immunoblotting analysis.

During up-scaled expression and purification, cells were induced in 400 ml of BMMY, 0.5% (v/v) methanol. The culture supernatant was i) collected after 72h by centrifugation, ii) filtered through 0.22 μm pore size and iii) dialysed and concentrated ~8x by ultrafiltration using a 5 MWCO Biomax membrane (Sigma-Aldrich) in Tris 50 mM, NaCl 300 mM, pH 8.0. Imidazole was added to a final concentration of 10 mM. Immobilized metal ion affinity chromatography was used for purification (HisTrap FF crude 5 ml column, Sigma-Aldrich). The column was equilibrated in Tris 50 mM, NaCl 300 mM, Imidazole 10 mM, pH 8.0. The supernatant was loaded. The column was washed in the same buffer with Imidazole 20 mM. Proteins were eluted in Tris 50 mM, NaCl 300 mM, Imidazole 500 mM, pH 8.0. Fractions were desalted in a HiPrep$^{TM}$ 26/10 desalting column and eluted in Phosphate Buffer Saline (PBS). Eluted proteins were i) submitted to proteolytic cleavage with thrombin (GE healthcare cat. no. 27-0846-01); ii) filtered through 30 MWCO Centricon$^®$ and iii) concentrated in 3 MWCO (Sigma-Aldrich).

## Synthetic DgDef1 without the CTPP domain (DgDef1*ΔSPΔCTPP*)

DgDef1*ΔSPΔCTPP* was chemically synthesized by conventional solid phase peptide synthesis yielding a purity of 95.54% reported by HPLC analysis (ProteoGenix SAS, France). Molecular weight correctness was confirmed by mass spectrometry. Lyophilized peptides were reconstituted in acetonitrile:water (1:3) to a final concentration of 12 mg/ml (2 mM), aliquoted and stored at -80˚C.

## SDS-PAGE and immunoblotting

Total protein content was quantified using Bovine Serum Albumin as a standard (Sigma Aldrich cat. no. B6916). Proteins were separated in 15% acrylamide gels using a Mini PRO-TEAN$^®$ electrophoresis system (Bio-Rad). Gels were revealed by silver staining [34]. For Immunoblotting, proteins were transferred to PVDF membranes in Towbin buffer [25 mM Tris (pH 8.3), 192 mM Glycine, 20% Methanol] at 20 V, 400 mA for 150 min (Mini Trans-Blot, Bio-Rad). Membranes were blocked in 4% non-fat dry milk and 2% BSA dissolved in Tris Buffer Saline pH 7.6 (TBS), then incubated overnight with 1:1000 monoclonal anti-poly-Histidine antibody (Sigma Aldrich cat. no. H1029) prepared in half strength blocking buffer diluted in TBS. This was followed by three washes in TBS, 0.1% Tween 20 (10 min each) and incubation for 1h in 1:5000 peroxidase-conjugated anti-mouse IgG produced in rabbit (Sigma

Aldrich cat. no. A9044) and three washes in TBS, 0.1% Tween 20 as before. The chemilumines-cence-based ECL^TM start Western Blotting kit RPN3243 was used for detection (Sigma-Aldrich). Images were captured in a Gel Doc^TM using the Quantity One® software v.4.5.2 (Bio-Rad).

## Effect of DgDef1 on differentiation of *Streptomyces coelicolor* A3(2) M145

These assays were performed with either DgDef1*ΔSP*, which lacks the N-terminal signal pep-tide, or with DgDef1*ΔSPΔCTPP*, which additionally lacks the acidic C-terminal domain. About $10^{-5}$ spores of *S. coelicolor* A3(2) M145 were plated on R5 agar [53] and 10 µl serial dilu-tions of a 9 µM peptide solution (in PBS) were spotted. 10 µl PBS was used as a negative con-trol. Alternatively, the plates were incubated at 30˚C for 24h to allow spore germination and growth of substrate mycelium before application of the peptide. After 3–7 days incubation at 30˚C, plates were checked for possible effects of the peptide on morphological differentiation.

For light microscopy of substrate mycelium, aerial mycelium, and spore chains, coverslips were inserted into LB or MS agar and inoculated with a diluted spore solution ($\sim 10^3$ spores) of *S. coelicolor* A3(2) M145 at the edge between the agar and inserted coverslips. Subsequently, 10 µl of 9 nM to 9 µM peptide solutions (in PBS) were pipetted into the gap between agar and coverslip. 10 µl PBS was used as a negative control. After 2 to 4 days of incubation at 30˚C, the coverslips were placed on microscope slides coated with a thin agar pad and 1 drop of phosphate-buffered saline (PBS). Images were captured using the phase-contrast mode of an Olympus BX60 microscope equipped with an Olympus UPlanFl 100× oil objective and an F-view II camera.

## Effects of DgDef1*ΔSPΔCTPP* and DgDef1*ΔSP* on *Sinorhizobium meliloti* 1021

*S. meliloti* 1021 was challenged during exponential growth in TY medium ($OD_{600} = 0.5$) with increasing amounts of the chemically synthesized DgDef1*ΔSPΔCTPP* (0.3, 1.3, 4.2, 8.3, 16.7, and 20.8 µM). Acetonitrile:water (1:3) was used as negative control. After the addition of DgDef1*ΔSPΔCTPP*, cultures (100 µl) were incubated at 30˚C, 150 rpm for 1h. To assess the number of surviving colony forming units (CFUs), cells were then diluted $10^{-6}$ and 40 µl were plated on TY plates containing 600 µg/ml streptomycin. In parallel, cell membrane integrity was accessed by fluorescence microscopy after staining with Propidium Iodide (PI; Sigma-Aldrich cat. no. P4170). Cells were collected by centrifugation, gently resuspended in Vincent Minimal Media (VMM; [54]) supplied with 20 ng/µl PI and incubated for 5 min in the dark; cells were collected again and gently resuspended in VMM. Cells were observed in a Nikon microscope Eclipse Ti-E equipped with a differential interference contrast (DIC) CFI Apoc-hromat TIRF oil objective (100x; numerical aperture of 1.49) and a phase-contrast Plan Apo l oil objective (100x; numerical aperture, 1.45) with the AHF HC filter set F36-504 for PI (ex bp 562/40 nm, bs 593 nm, and em bp 624/40 nm). Images were acquired with an Andor iXon3 885 electron-multiplying charge-coupled device (EMCCD) camera.

In a similar fashion, *S. meliloti* 1021 cells growing exponentially ($OD_{600} = 0.48$) were chal-lenged with either 2.5 µM of DgDef1*ΔSP* or with PBS (control); cultures (100 µl) were incubated at 30˚C, 150 rpm for 1h. To assess the survival rate (CFUs), cells were then diluted $10^{-6}$ and 30 µl were plated on TY agar containing 600 µg/ml streptomycin. Cultures were maintained at 30˚C until colonies appeared and could be counted. Statistics were calculated in RStudio [42].

## RNA isolation and RNAseq from *Sinorhizobium meliloti* 1021

In order to obtain sufficient RNA for RNAseq, the previously described growth set up was up-scaled. *S. meliloti* 1021 ($OD_{600} = 0.5$) was challenged in 10 mL TY medium with 25 µg/mL of

DgDef1*ΔSPΔ*CTPP in 50 mL conical tubes at 30˚C, 150 rpm for 1h. The negative control was challenged with acetonitrile:water (1:3). Assays were performed in triplicate. For RNA isolation, cells were i) precipitated (7.800 x *g*, 5 min); ii) resuspended in 1 mL of QIAzol lysis reagent; iii) and homogenized in a FastPrep® sample preparation system (3 x 6.500 rpm, 20 sec, 15 sec break) using Lysing Matrix B containing 0.1 mm silica beads (MP biomedicals, cat. no. 6911). Suspensions were then i) incubated 5 min at 15–25˚C and 140 μl of chloroform was added; ii) shaken vigorously for 15 sec and reincubated for 3 min at 15–25˚C and iii) centrifuged (11.300 x *g*, 15 min, 4˚C) to collect the upper aqueous phase, which was subsequently mixed with 1.5 volumes of absolute ethanol. The protocol proceeded with the miRNeasy Mini Kit (Qiagen, cat. no. 217004) according to the manufacturer's instructions with an on-column DNase digestion. Since only a single library was prepared for each condition (*i.e.*, treated and untreated), attention was paid to include RNA representing the four independent biological replicates in order to minimize the possibility of results biased by sample choice. Pooled samples used for RNAseq contained equal amounts of RNA from each biological replicate. rRNA depletion was conducted with the Illumina Ribo-Zero rRNA removal Kit (Bacteria). The cDNA library was prepared using the NEB Ultra directional Kit and sequencing was performed on an Illumina HiSeq3000 platform. 6,979,921 and 6,650,320 reads were obtained for the (pooled) untreated and the (pooled) treated samples, respectively. 99.64% and 99.67% of the sequencing reads could be mapped to the *S. meliloti* 1021 reference genome using Bowtie v.1.2.3. Differential gene expression analysis was performed with R scripts (DESeq2); sequences are available at https://www.ebi.ac.uk/arrayexpress/ (accession number E-MTAB-11181). For gene expression analysis by RT-qPCR, RNA from individual replicates was used.

## Results

### Nodule-specific cysteine-rich peptides of all nodulating plants have a common origin

During the search for nodule-specific genes of *D. glomerata*, Demina and collaborators identified two genes encoding defensin-like peptides, *DgDef1* and *DgDef2*. In both cases, the defensins contained an acidic C-terminal domain [28]. After increasing the sequencing depth, four more genes encoding defensin-like peptides were identified in nodules of *D. glomerata*, namely *DgDef3*, *DgDef4*, *DgDef5*, and *DgDef6* [29]. The defensin domains of these peptides contain a series of highly conserved cysteine residues (S2 Fig). The products of *DgDef3* and *DgDef4* possess an acidic C-terminal domain like DgDef1 and DgDef2 (S2 Fig). All six peptides possess a signal peptide (SP) for synthesis and uptake in the endomembrane system.

The phylogeny of these cysteine-rich domains was inferred using defensins and cysteine-rich peptides from non-symbiotic plant species as well as nodule-specific cysteine-rich peptides (NCRs) from legumes and nodule-specific defensins from actinorhizal plants [26,27,29]. The resulting tree (Fig 1) showed that the NCRs of legumes and the defensins of actinorhizal species are part of the same relatively well-supported clade (80% bootstrap, see star in Fig 1). This clade included nodule-specific defensins from all orders of symbiotic plants: Fabales (legumes [26]), Fagales (*Alnus glutinosa* and *Casuarina glauca* [27]), Rosales (*Ceanothus thyrsiflorus* [28]) and Cucurbitales (*D. glomerata*) (Fig 1).

### Genes encoding defensin-like peptides are highly expressed in nodules of *Datisca glomerata* compared to roots

Quantitative Reverse Transcription-Polymerase Chain Reaction (RT-qPCR) analysis showed that all the genes encoding members of the *DgDef* family with a CTPP were highly expressed

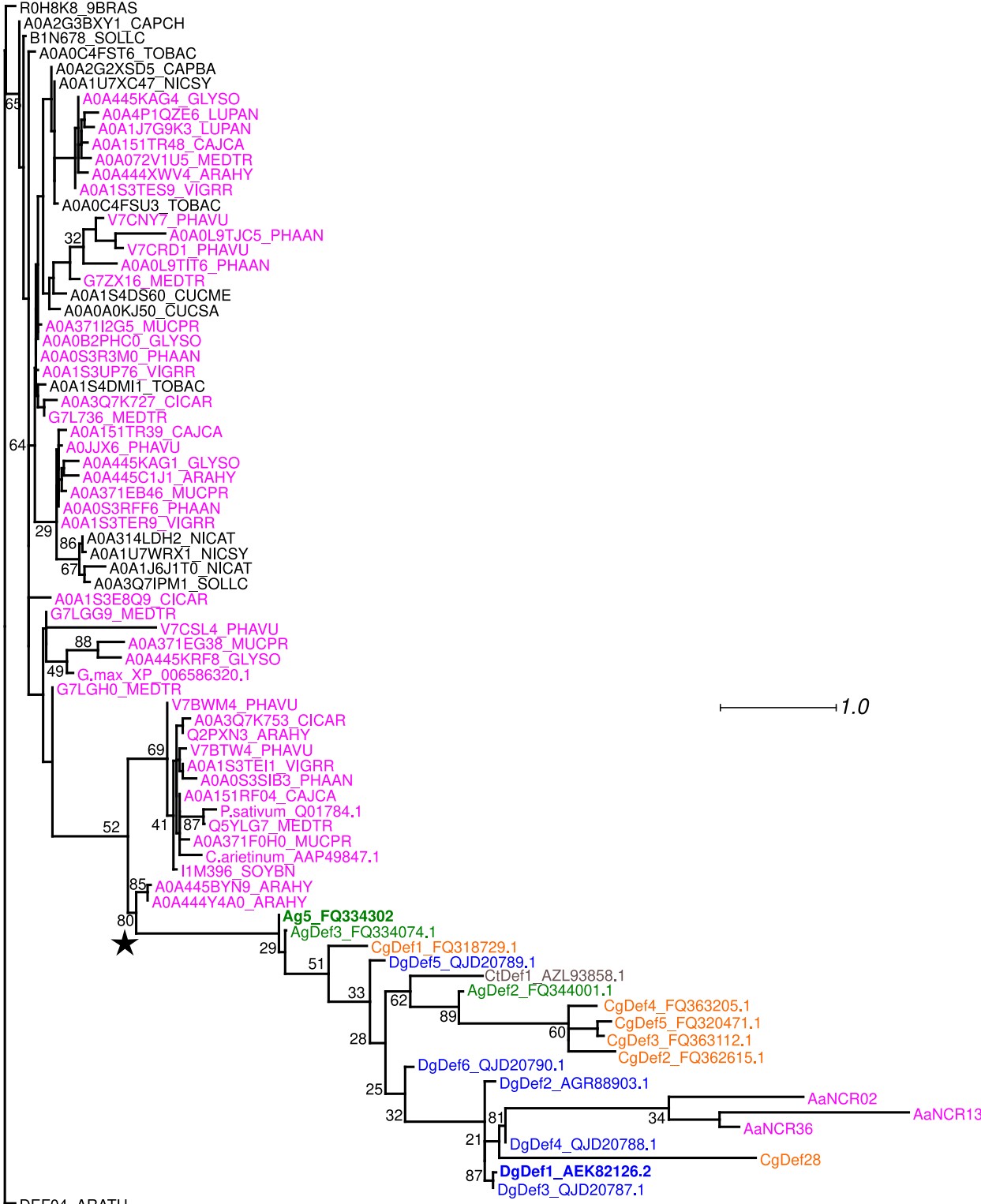

**Fig 1. Phylogeny of _Datisca glomerata_ defensin-like peptides.** Maximum-likelihood (ML) phylogenetic tree based on the cysteine-rich domains that are presumably linked with antimicrobial activity. The tree was rooted with the _Arabidopsis_ defensin DEF04_ARATH. Colour attributes: Legumes, Fabales (magenta), actinorhizal species such as _Alnus glutinosa_ (green), _Casuarina glauca_ (orange), _Ceanothus thyrsiflorus_ (grey), and _Datisca glomerata_ (blue). _D. glomerata_ peptides containing a C-terminal acidic domain are DgDef1, DgDef2, DgDef3, and DgDef4 (detailed in S2 Fig). The previously characterized _A. glutinosa_ peptide, Ag5 [27], is given in green bold print. Nomenclature is mainly from UniProtKB/Swiss-Prot,

otherwise from the GenBank and EMBL databases. Sequences from members of the Fagales (*Alnus* and *Casuarina*) are based on Carro et al. [37] and sequences from *Aeschynomene* are based on Czernic et al. [26]. A star labels the subclade of legume NCRs and actinorhizal nodule-specific defensins. Scale bar represents the ML estimate of the average number of substitutions per site between two nodes. ML bootstrap support is given for each branch.

in nodules compared to roots (Fig 2). The family member expressed at the highest level in nodules, *DgDef1*, was selected for further analysis.

## The CTPP domain of DgDef1 does not serve as a vacuolar targeting signal in tobacco

A negatively charged C-terminal domain has been identified in some other defensins, particularly in defensins from members of the Solanaceae [25]. These domains, which do not share sequence similarity with the CTPPs of DgDef1, DgDef2, DgDef3 and DgDef4, had been shown to represent a vacuolar targeting signal. In spite of the lack of homology, for safety we examined whether the CTPP of DgDef1 could represent a vacuolar targeting signal. Regions of the DgDef1 ORF were fused with the green fluorescent protein (GFP) ORF (schematic details in S1A Fig) and cloned in a binary vector for *Agrobacterium tumefaciens*-mediated transient expression in leaves of *Nicotiana benthamiana*. To identify vacuolar expression, results of plasmolyzed and non-plasmolyzed leaves were compared. The results are depicted in Fig 3. A construct that contained the SP of DgDef1 fused to GFP led to green fluorescence in the apoplast (Fig 3A and 3C). Similarly, a construct where GFP was fused with the SP of DgDef1 at the N-terminus and the CTPP at the C-terminus, led to GFP fluorescence in the apoplast (Fig 3B and 3D). Without added domains of DgDef1, GFP located to the nucleus (Fig 3E). Thus, the CTPP domain of DgDef1 is not involved in targeting the peptide to the vacuole.

## *DgDef1* is expressed in young infected cells and transiently in the nodule lobe meristem

The DgDef1 promoter was amplified and sequenced using the genome walking method (NCBI accession number MZ779183) and fused to the β-glucuronidase ORF for analyzing the expression pattern in nodules formed on hairy root induced by *Agrobacterium rhizogenes*. Five different transformations were performed, the first two with the method established by Markmann et al. [48], then with a modified method [49]. In the first two experiments, GUS staining denoting activity of the DgDef1 promoter was found in young infected cells (Fig 4A) and, transiently, at the apex of incipient nodule lobes that did not yet contain infected cells (Fig 4B). In the transformations using the new method, however, while expression at the tips of nodule lobes was still detected, expression in young infected cells was not (Fig 4C and 4D). Sequencing of the promoter showed that it had acquired mutations since the previous experiments (S3 Fig).

## DgDef1 does not affect hyphal growth and differentiation of Gram-positive *Streptomyces coelicolor*

To investigate the activity of DgDef1 against bacteria, isolated peptide was required. The yeast *Pichia pastoris* was used to produce a recombinant peptide, coined DgDef1ΔSP (details in S1B Fig). The purification yielded a total of 140 μg of DgDef1ΔSP in high purity; the produced peptide turned out to be prone to form strong multimeric structures (S4 Fig). Oligomeric structures seem to be a feature of anti-bacterial peptides as they were previously observed in other plant defensins [55–58].

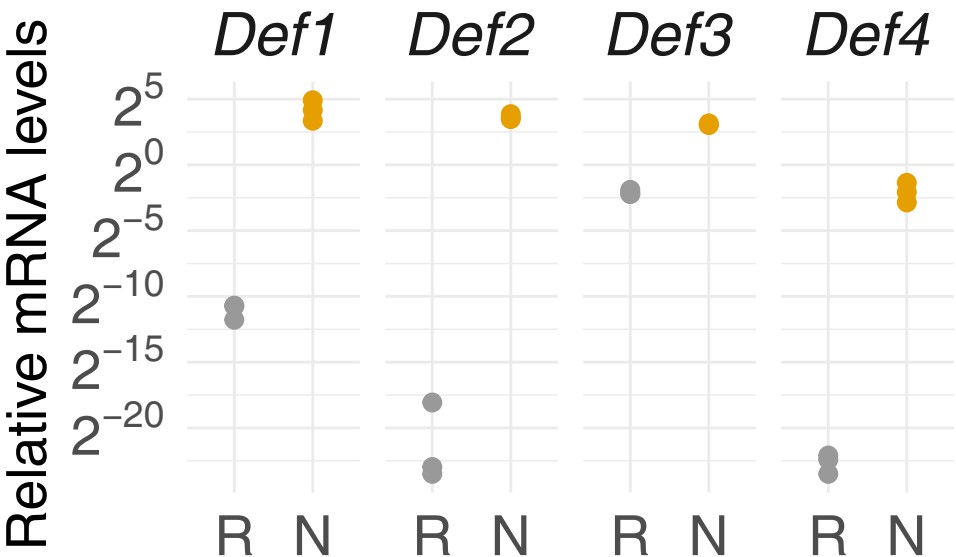

**Fig 2. Relative transcript abundance of genes encoding a subfamily of defensin-like peptides in subterranean organs of *Datisca glomerata*.** Expression levels in roots (R; grey dots) and nodules (N; orange dots) are relative to those of the housekeeping gene *EF-1A* (n = 3). Significance of differences between R and N was at $p<0.01$ for all genes examined based on Student's t-test. *Y*-axis is given in $\log_2$ scale.

Since the microsymbionts of *D. glomerata* are Gram-positive, the inhibitory effect of DgDef1ΔSP on the model actinobacterium *Streptomyces coelicolor* A3(2) was examined. *Streptomyces coelicolor* A3(2) is particularly suitable as a reporter of inhibitory effects, since even sub inhibitory concentrations of compounds, which do not inhibit mycelial growth often affect morphological differentiation or interfere with the production of pigmented antibiotics. Spores of *Streptomyces coelicolor* A3(2) M145 were plated on R5 agar and challenged with a serial dilution of DgDef1ΔSP, ranging from 9 nM to 9 μM. No effects of the peptide on growth nor on morphological differentiation were observed. The possible effects of DgDef1ΔSP on the different stages of the life cycle of *S. coelicolor* A3(2) M145 were addressed by phase contrast microscopy. However, neither vegetative growth by apical tip extension and branching, nor septation of aerial mycelium and formation of proper spore chains were affected by the presence of DgDef1ΔSP. This outcome raised the question whether the presence of the CTPP could interfere with the activity of DgDef1ΔSP; it could not be excluded that the CTPP had to be first cleaved off for the peptide to become active. To address this question, the 51 residues encompassing the defensin domain of DgDef1 were chemically synthesized and the synthetic peptide DgDef1ΔSPΔCTPP was then used to repeat the experiments with *S. coelicolor* A3(2) M145 as described above. However, these assays led to the same outcome as those performed with DgDef1ΔSP. In summary, neither DgDef1ΔSP nor DgDef1ΔSPΔCTPP could affect the growth and/or differentiation of *S. coelicolor* A3(2) M145.

## DgDef1ΔSPΔCTPP acts as an antimicrobial peptide against Gram-negative *E. coli* K-12 substrain MG1655 and *Sinorhizobium meliloti* 1021

To investigate whether the synthetic DgDef1ΔSPΔCTPP instead had an effect on Gram-negative bacteria, *E. coli* K-12 substrain MG1655 and *S. meliloti* 1021 were challenged with a range of concentrations of DgDef1ΔSPΔCTPP, and bacterial growth in presence and absence of DgDef1ΔSPΔCTPP was quantified. During the pilot assay, DgDef1ΔSPΔCTPP showed a similar negative effect on the growth of both strains (S5 Fig). Because of its role in root nodule

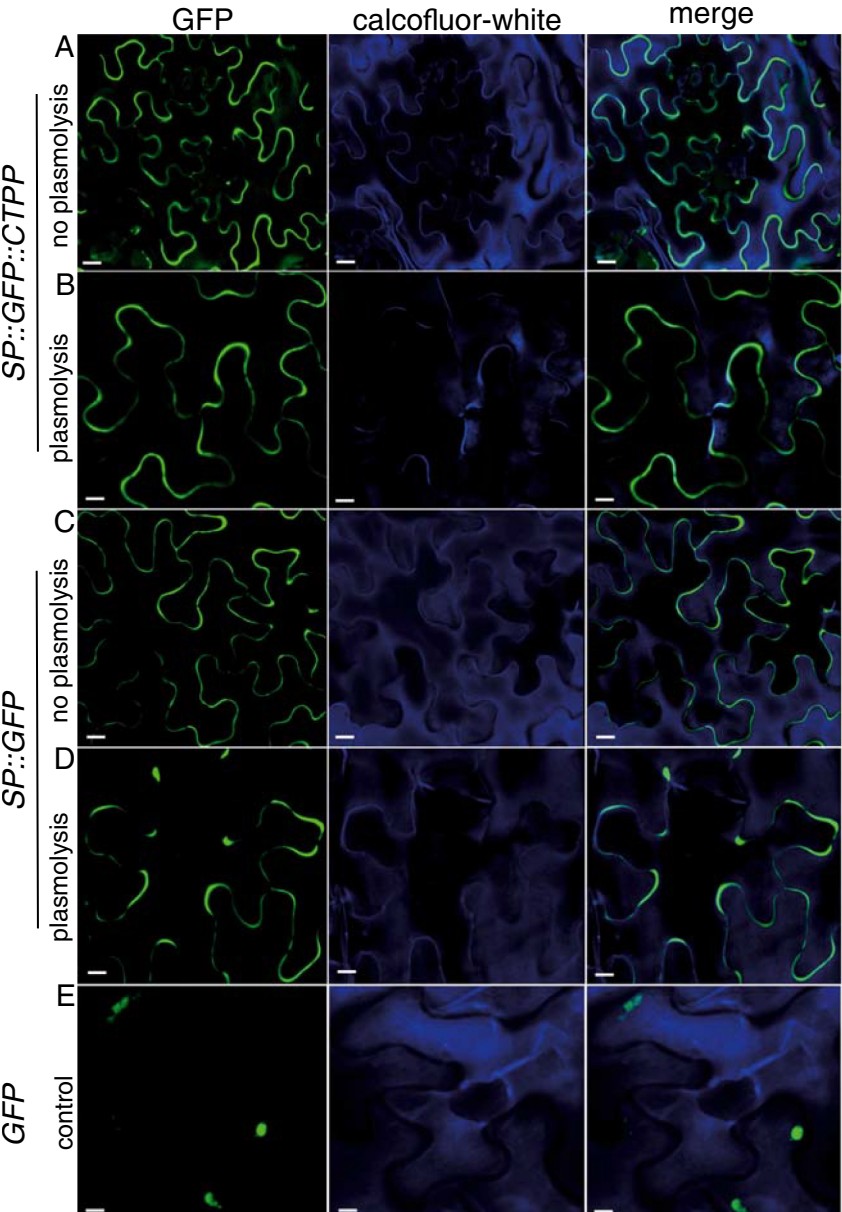

**Fig 3. Subcellular confocal localization of GFP after *Agrobacterium tumefaciens*-mediated transient expression of chimeric cDNAs in tobacco.** The N-terminus of GFP was fused to the native signal peptide SP of DgDef1, and the C-terminus was fused (A, B) or not (C, D) to the C-terminal domain of DgDef1, CTPP. Shown are confocal laser scanning microscopy images in non-plasmolyzed (A, C, E) and plasmolyzed (B, C) tissues with excitations at 488 nm (GFP), 405 nm (calcofluor-white), or both (merge). Panel E displays GFP alone. Chimeric proteins and light conditions are given. Scale bars: A, 5 μm; B, 4 μm; C, 3.5 μm; and D, E 3 μm.

symbioses, *S. meliloti* was chosen for further experiments. To define the minimal inhibitory concentration of DgDef1ΔSPΔCTPP that could exert an effect on *S. meliloti*, a growth curve was traced based on increasing concentration of peptide. Results from three independent experiments showed that 50 μg/ml (8.3 μM) of DgDef1ΔSPΔCTPP were sufficient to reduce *S. meliloti* growth by 30%, when compared to its untreated control (p<0.001). Surprisingly, the effect of 100 μg/ml DgDef1ΔSPΔCTPP on the growth of *S. meliloti* growth was not significantly different from that of 50 μg/ml, while 125 μg/ml (20.8 μM) of DgDef1ΔSPΔCTPP

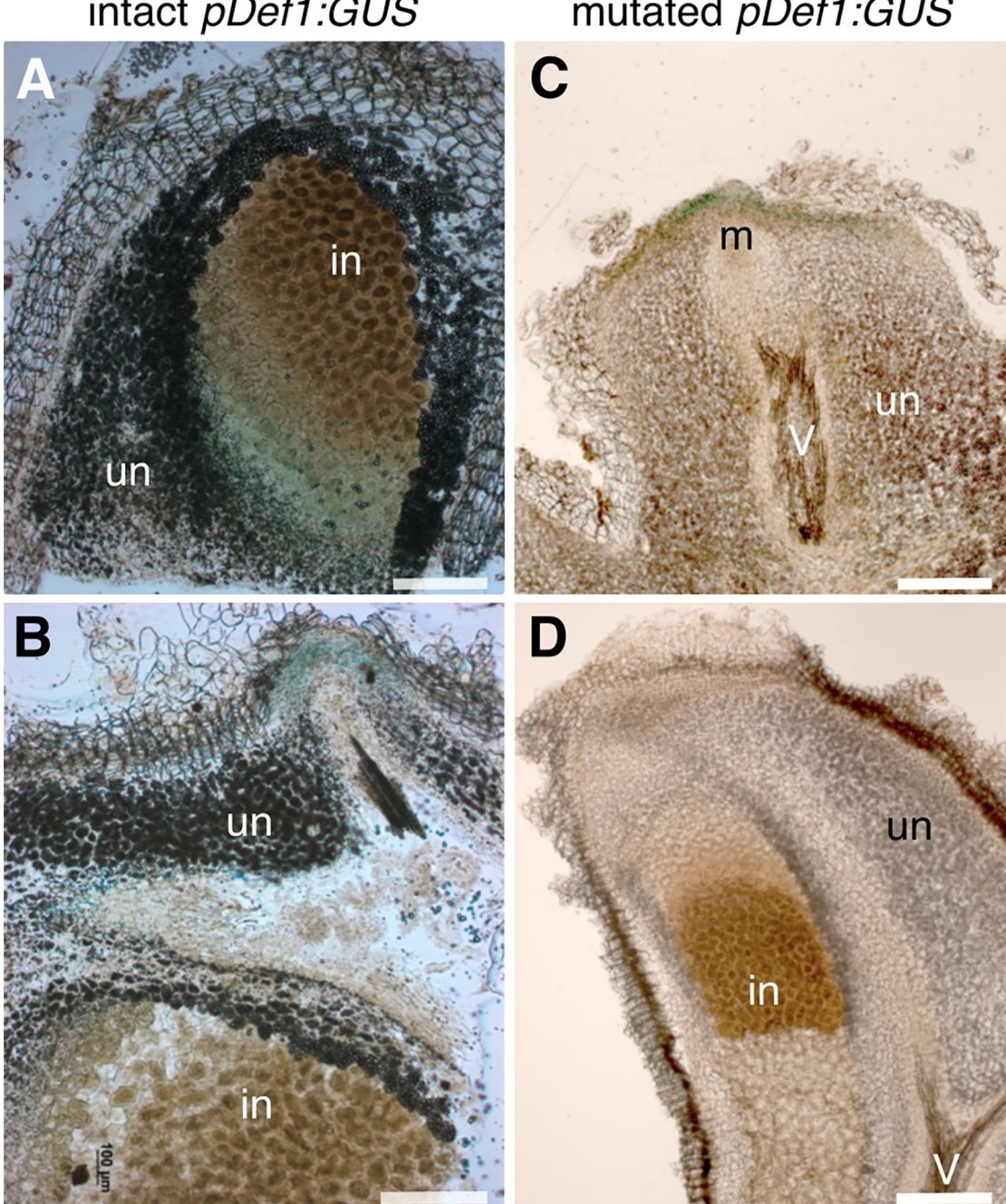

**Fig 4. Expression of *DgDef1:GUS* in transgenic hairy roots of *Datisca glomerata*.** Light micrographs of nodule sections are shown. Transformation with wild type **(A, B)** and mutated **(C, D)** promoter *DgDef1:GUS*. Panel **(A)** shows GUS activity in the young infected cells which are not yet filled with branching *Frankia* hyphae (infected cells, "in"), but not in the uninfected cells filled with starch grains (uninfected cells, "un"). **(B)** shows the transient GUS activity above the meristem of a nodule lobe. **(C)** shows again the transient GUS activity above the meristem of a nodule lobe, this time driven by the mutated *DgDef1* promoter, which **(D)** does not direct GUS expression in the young infected cells. V, vascular system. Size bars denote 300 μm.

reduced the growth of *S. meliloti* by 50% (IC$_{50}$; Fig 5A). These observations were supported by life/dead staining microscopy performed in parallel, showing that DgDef1ΔSPΔCTPP cytotoxicity led to membrane disruption (Fig 5B).

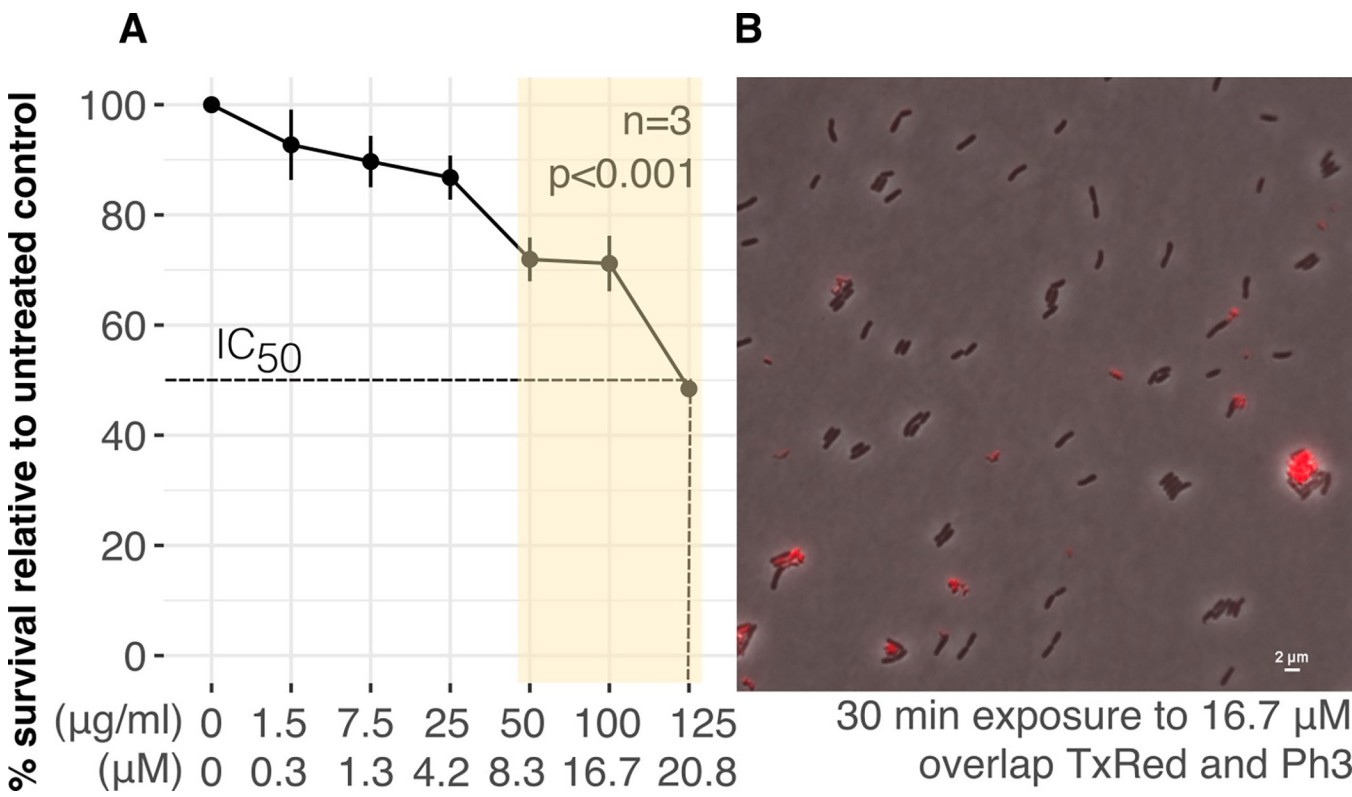

**Fig 5. DgDef1ΔSPΔCTPP activity towards *Sinorhizobium meliloti* strain 1021.** Cell survival is shown as colony forming ability **(A)** and membrane integrity **(B)**. The shadowed area in **(A)** covers the concentration range to which significance was assigned by Poisson binomial regression (p<0.001). Peptide concentration array is indicated. Bars depict standard deviation of three independent experiments (n = 3). **(B)** shows Propidium Iodide (PI) life/dead staining in an overlap of phase contrast (Ph3) with the filter set F36-504 at 593 nm for PI (TxRed).

### The CTPP domain does not impair the activity of DgDef1 towards *Sinorhizobium meliloti* 1021

To address the question whether the cleavage of the CTPP domain could be a requirement for the activity of DgDef1 towards *S. meliloti* 1021, exponentially growing cells were challenged with the *Pichia*-produced DgDef1ΔSP. Results showed that DgDef1ΔSP could affect *S. meliloti* viability significantly (p = 0.002) when cells were exposed to a peptide concentration as low as 2.5 μM, when compared to the control (Fig 6).

### Analysis of DgDef1-induced transcriptional changes in *Sinorhizobium meliloti* 1021 by RNAseq and RT-qPCR

To access the global transcriptome responses of exponentially growing *S. meliloti* to 1 h challenge with 4.2 μM of DgDef1ΔSPΔCTPP, two Illumina RNAseq libraries were prepared (treated *vs.* untreated). Only genes displaying at least a twofold induction or suppression were considered for comparison between libraries. In total, 284 genes (representing 4.5% of the predicted coding sequences in *S. meliloti 1021*) showed differential regulation; however, most of these genes had low expression levels (S2 Table). Genes that showed differential expression and high expression levels were selected for analysis by RT-qPCR, *SMc01242* (signal peptide DUF1775 domain containing protein), *SMc01800* (cytochrome C oxidase assembly protein subunit 15), *SMc02357* (high affinity ABC transporter for branched-chain amino acids, ATP-binding protein), *SMc02255* (*qtxA/cydA*, encoding Cytochrome d ubiquinol oxidase subunit I)

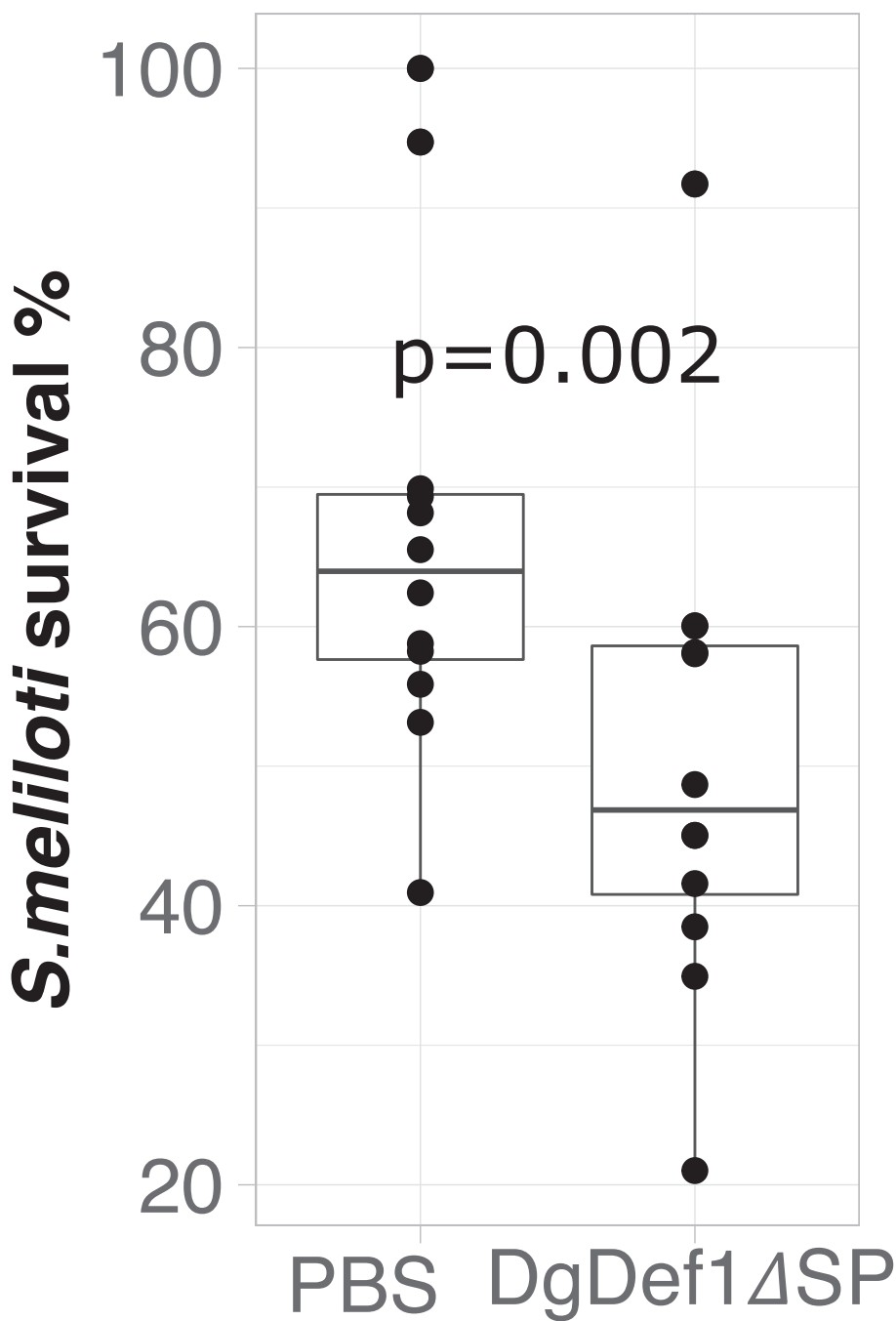

**Fig 6. DgDef1ΔSP activity towards *Sinorhizobium meliloti* strain 1021.** Cell survival is shown as percentage (%) in colony forming ability after 1 h treatment with PBS (control) or with 2.5 μM of DgDef1ΔSP. Significance of differences (p = 0.002) between conditions was calculated using a paired Student's t-test.

and *SMb21487* (*cyoA*, encoding cytochrome o ubiquinol oxidase chain II). The results are shown in Fig 7. Expression levels of *cyoA* and *qtxA/cydA* were significantly enhanced in response to treatment with DgDef1ΔSPΔCTPP (p<0.005). It is interesting that the expression

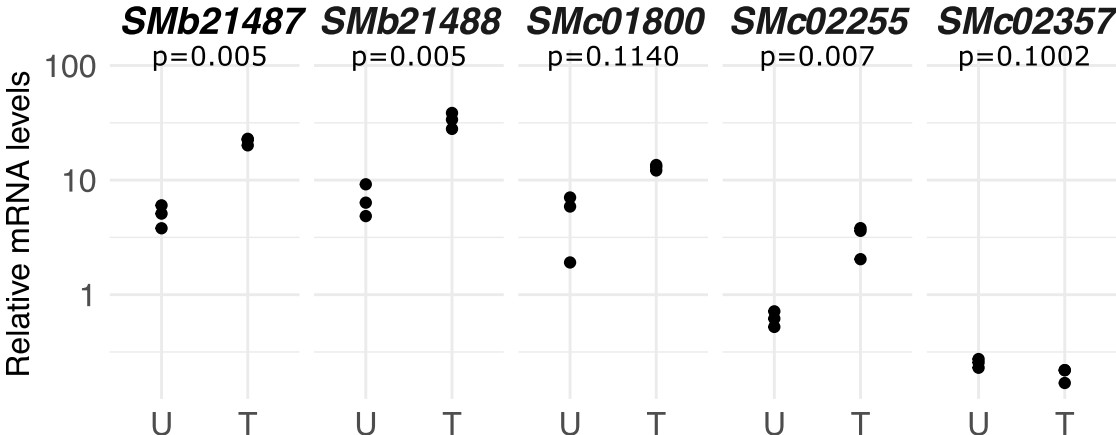

**Fig 7. Relative transcript abundance of *Sinorhizobium meliloti* 1021 genes in response to treatment with a sublethal concentration of DgDef1ΔSPΔCTPP.** Expression levels in untreated (U) and cultures treated with 4.2 μM of DgDef1ΔSPΔCTPP (T) are relative to those of the housekeeping gene *SMc01852* (n = 3). *p*-values were calculated using Student's t-test and are indicated on top of the panels. *Y*-axis is given in $\log_{10}$ scale.

of *SMc02357* and of *SMc01800* was also enhanced tendentiously, though below the level for statistical significance (p = 0.11), as the uptake of branched chain amino acids is a bacteroid feature due to the symbiotic auxotrophy of rhizobia for branched chain amino acids [59].

## Discussion

Phylogenetic analysis of the cysteine-rich domain of defensins from multiple symbiotic and non-symbiotic plant species, including legume NCRs and actinorhizal nodule-specific defensins (ANDs) showed that NCRs and ANDs are part of a distinct, monophyletic subclade of defensins (asterisk in Fig 1). This finding suggests many cases of convergent evolution; however, a common evolutionary origin of the nodule-specific cysteine-rich peptides in legumes and actinorhizal plants would seem more parsimonious. This would imply that while nodule-specific defensins occur across the different orders of actinorhizal plants [28], NCRs were actually lost in most legume lineages, instead of having evolved in a few of them.

Legume NCRs induce terminal differentiation in symbiotic rhizobia which involves endoreduplication, inhibition of cell division, increased membrane permeability and basically abolishes bacterial reproduction [21]. No terminal differentiation has been observed in symbiotic *Frankia*. The induction of terminal differentiation by NCRs has been suggested to represent legumes' strategy against rhizobial 'cheaters' [60]). The existence of rhizobial 'cheaters' has been proposed as one of the reasons why root nodule symbioses were counter-selected during evolution [13,14,61]. In this context, it is surprising that NCRs are found only in two groups of legumes [62]. Hence, if NCRs were lost in most legume lineages, it seems that terminal differentiation not only affects the reproductive success of 'cheaters', but also that of efficient rhizobial symbionts, and therefore was counter-selected.

In any case, a subfamily of ANDs with an acidic C-terminal propeptide (CTPP) domain evolved in *D. glomerata*. The option that the CTPP domain is acting as a signal in vacuolar targeting is not supported by this study (Fig 3), however the possibility that CTPP acts as a targeting signal to the perisymbiont space cannot be excluded. A synthetic version of DgDef1 lacking its CTPP domain (DgDef1ΔSPΔCTPP) could affect the growth of *E. coli* K12 substrain MG1655 as well as that of *Sinorhizobium meliloti* 1021 in culture. This result supports the assumption that the CTPP domain is not required for the cytotoxic effect of DgDef1, at least

concerning these Gram-negative bacterial strains. However, since no effect was observed towards *Streptomyces coelicolor*, independent of the presence or absence of the CTPP domain, the question about the role of the CTPP domain in nodules of *D. glomerata* remains open. Different options exist: the CTPP domain may be either acting i) on plant's self-protection, *i.e.*, protecting the plant cytosol from the cytotoxic effects of DgDef1; ii) in subcellular targeting to the perisymbiont space; iii) on binding to another globular protein.

The latter option raises further questions since such interactions may rely on factors such as pH, redox potential, or post-translational modifications. In this context, intrinsically disordered protein domains carry out important biological functions involving protein/ligand interactions [63]. IUPred2A (https://iupred2a.elte.hu/), a software that predicts protein disorder as a function of redox state and binding properties [63], predicted for DgDef1 a state of disorder above the established threshold for the region comprising the CTPP domain (S6A Fig). These results were also supported by Anchor2, an algorithm that recognizes disordered binding regions ([64]; S6A Fig). In addition, prediction of the redox disorder for the *Pichia*-produced peptide (DgDef1ΔSP) showed a high disorder score, including the region spanning the CTPP residues (S6B Fig). On the other hand, it is tempting to speculate that the CTPP domain is responsible for targeting DgDef1 towards intracellular microsymbionts, which would imply that the targeting processes differ in actinorhizal Cucurbitales compared to actinorhizal Fagales and Rosales. This assumption is consistent with and provides cues to distinct growth of the persistent infection threads harboring the intracellular microsymbionts in Cucurbitales, which differs from actinorhizal Fagales and Rosales [65].

The effect of the defensin domain of DgDef1 (DgDef1ΔSPΔCTPP) on bacteria could not be analysed with the microsymbionts of *D. glomerata* as they cannot be cultured [31,32]. Instead, it was analysed for a well-characterized Gram-positive strain, *Streptomyces coelicolor* A3(2) M145, and two equally well-characterized Gram-negative strains, *E. coli* K-12 substrain MG1655 and *S. meliloti* 1021. No effect on the growth and differentiation of *S. coelicolor* A3(2) M145 could be detected. However, DgDef1ΔSPΔCTPP inhibited the growth of *E. coli* K-12 substrain MG1655 as well as that of *S. meliloti* 1021. Detailed analyses with *S. meliloti* 1021 showed an $IC_{50}$ of 20.8 μM. This $IC_{50}$ is higher than that displayed by other antibacterial defensins, which can have $IC_{50}$s as low as 0.1 μM [5], and it is also higher than the $IC_{50}$ of some legume NCRs, which can be as low as 5 μM [22].

An analysis of the effect of a sublethal concentration (4.2 μM) of the defensin domain of DgDef1 (DgDef1ΔSPΔCTPP) on the *S. meliloti* 1021 transcriptome showed one clear difference with the effect of *Medicago truncatula* NCRs: the defensin domain of DgDef1 did not reduce the expression of the cell cycle regulator *ctrA* [66]. This should not surprise in that there is no evidence for cell cycle control by the plant in actinorhizal symbioses, and it is unclear how the endoreduplication induced in unicellular bacteroids by legume NCRs would affect a mycelial bacterium. Apart from that, the defensin domain of DgDef1 enhanced the expression levels of the *cyoABC* operon located on pSymB, which encodes a cytochrome *o* ubiquinol oxidase, a low $O_2$ affinity oxidase with a high proton pumping activity that is induced following a shift to acidic pH [67], which also happens in symbiosis as the peribacteroid space is acidified during bacteroid differentiation [68]. This ubiquinol oxidase is also involved in the shift from aerobic to anaerobic growth in *E. coli* [69]. It was also found to be induced in *Pseudomonas aeruginosa* by $H_2O_2$ treatment [70] and in *tolC* mutants of *S. meliloti* 1021, suggesting an involvement in the response to oxidative stress. Similarly, the expression of *qtxA/cydA*, which encodes a cytochrome d ubiquinol oxidase subunit, was enhanced in response to DgDef1 treatment. The expression of both the *cyoABC* operon and of *qxtA/cydA* is enhanced in *S. meliloti* in response to iron limitation or when the *rirA* gene, which controls the response to iron limitation, is mutated [71]. Both *qtxA/cydA* and *cyoAB* are relevant in symbiosis; based

on Roux et al. [72], *cyoAB* show substantial expression in all zones of the nodule inner tissue, while *qxtA* displayed the highest expression levels in the zone of nitrogen fixation. In short, while DgDef1 has a cytotoxic effect on *S. meliloti* 1021, at sublethal concentrations it induced some changes in the expression of genes related to energy metabolism that are compatible with the reaction to oxidative stress, and are also compatible with the changes in metabolism that occur during bacteroid differentiation.

## Conclusions

Legumes as well as actinorhizal plants evolved cysteine-rich peptides expressed in infected nodule cells from the same subclade of defensins; while these peptides were found in all acti-norhizal plants examined, they are missing in nodules of most legumes. In nodules of *Datisca glomerata*, *DgDef1* is expressed transiently in the nodule meristem during nodule induction and later in young infected cells. Without both its signal peptide and its acidic C-terminal domain, DgDef1 has a cytotoxic effect on two different Gram-negative bacterial strains tested, but did not affect the growth of a *Streptomyces* strain. At sublethal concentrations, it induces the expression of terminal quinol oxidases in *Sinorhizobium meliloti* 1021; these oxidases are involved in the oxidative stress response and also expressed in symbiosis. Taken together, the nodule-specific defensin from an actinorhizal member of the Cucurbitales had effects on a rhizobium strain and the induced changes resemble that of legume NCRs.

## Supporting information

**S1 Fig.** Schematic presentation of constructs used for subcellular localization (A) and heterologous production (B) of DgDef1ΔSP. Panel A displays the three GFP chimeras generated for *Agrobacterium tumefaciens*-mediated transformation of *Nicotiana benthamiana* (see Materials and Methods). Panel B illustrates the strategy employed to prepare a synthetic cassette for expression of DgDef1ΔSP in *Pichia pastoris*. SignalP (v.4.1) predicted a signal peptide (SP) in the DgDef1 ORF (see plot). Taking advantage of pIC9K as a secretion vector, the native DgDef1 SP was replaced by the synthetic SP located downstream of the strong promoter AOX1. A 6-His-tag and a thrombin cleavage site were engineered (see Materials and Methods). Note: Depicted domains are not on scale.
(TIFF)

**S2 Fig. Amino acid sequence alignment of members of a distinct family of defensin-like peptides formed in nodules of *Datisca glomerata*.** Multiple sequence alignment of six peptides is shown. The signal peptide cleavage site is indicated by a vertical dashed line. Note the presence of the characteristic CTPP domain in four members of the family.
(EPS)

**S3 Fig. Point mutations in the *DgDef1* promoter that led to loss of expression in young infected cells.** Numbers denote distance from the ATG (A = +1) of the *DgDef1* ORF. Differences between the original DgDef1 promoter and the mutated version are highlighted in yellow. The TATA box is highlighted in light blue. Homology is indicated with an 38 bp stretch of the 200 bp region of the pea *ENOD12* promoter that is sufficient for nodule-specific and Nod factor-induced expression (Vijn et al., 1995); asterisks indicate nucleotide conservation.
(EPS)

**S4 Fig. Purification of DgDef1ΔSP overexpressed in *Pichia pastoris*. (A)** Chromatogram showing the elution peak of DgDef1ΔSP. **(B)** Immunoblotting analysis summarizing the different purification steps. **(C)** Purification table. **(D)** Silver staining showing oligomeric structures

of purified DgDef1ΔSP.
(TIFF)

**S5 Fig. Preliminary assays to test the viability of *Sinorhizobium meliloti* 1021 and its *bacA* mutant when challenged with DgDef1ΔSP.** Time of exposure and peptide concentrations are indicated.
(EPS)

**S6 Fig. Biophysical predictions of intrinsically disordered regions of DgDef1. (A)** The predicted output of IUPred2 (red) and ANCHOR2 (blue) for DgDef1. **(B)** Redox-state-dependent IUPred2 prediction for the *Pichia*-produced peptide DgDef1ΔSP. The estimated sensitivity of the disorder tendency is marked in purple.
(TIFF)

**S1 Table. Primers used in this study.**
(XLSX)

**S2 Table. *Sinorhizobium meliloti* 1021 genes whose expression levels were changed significantly by treatment with sublethal concentrations of DgDef1ΔSPΔCTPP (RNAseq analysis).**
(XLSX)

# Acknowledgments

The authors would like to thank Peter Lindfors and Anna Pettersson (Stockholm University) for taking care of the *D. glomerata* and *N. benthamiana* plants, Robert Benezra (Sloan Kettering Institute) for the gift of plasmid H2-Venus, Max Griesmann (LMU Munich) for providing a new transcriptome assembly for completing the ORFs of *DgDef3* and *DgDEF4*, and Doreen Meier (Philipps-Universität Marburg) for handling the RNA samples for sequencing and public data reposition. Confocal microscopy was performed at the Imaging Facility of Stockholm University (IFSU).

# Author Contributions

**Conceptualization:** Marco Guedes Salgado, Katharina Pawlowski.

**Formal analysis:** Marco Guedes Salgado, Irina V. Demina, Anke Becker.

**Funding acquisition:** Anke Becker, Katharina Pawlowski.

**Investigation:** Marco Guedes Salgado, Irina V. Demina, Pooja Jha Maity, Anurupa Nagchowdhury, Elizaveta Krol, Günther Muth, Anke Becker.

**Methodology:** Marco Guedes Salgado, Irina V. Demina, Pooja Jha Maity, Anurupa Nagchowdhury, Andrea Caputo, Elizaveta Krol, Christoph Loderer, Günther Muth, Anke Becker.

**Supervision:** Pooja Jha Maity, Anke Becker, Katharina Pawlowski.

**Visualization:** Marco Guedes Salgado, Irina V. Demina, Pooja Jha Maity.

**Writing – original draft:** Marco Guedes Salgado, Irina V. Demina.

**Writing – review & editing:** Marco Guedes Salgado, Irina V. Demina, Pooja Jha Maity, Anurupa Nagchowdhury, Andrea Caputo, Elizaveta Krol, Christoph Loderer, Günther Muth, Anke Becker, Katharina Pawlowski.

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
