## [Decision Letter · Decision Letter 0]

7 Dec 2021

PONE-D-21-31806DgDef1, a defensin formed specifically in infected cells of root nodules of the actinorhizal plant Datisca glomerata, induces membrane disruption in Sinorhizobium meliloti 1021PLOS ONE

Dear Dr. Pawlowski,

Thank you for submitting your manuscript to PLOS ONE. After careful consideration, we feel that it has merit but does not fully meet PLOS ONE’s publication criteria as it currently stands. Therefore, we invite you to submit a revised version of the manuscript that addresses the points raised during the review process.Specifically, look into the Title, materials and methods, discussion and conclusion section besides detailed comments by the reviewers.

We look forward to receiving your revised manuscript.

Kind regards,

Dharam Singh

Academic Editor

PLOS ONE

Journal Requirements:

The authors would like to thank Peter Lindfors and Anna Pettersson (Stockholm University) for taking care of the D. glomerata and N. benthamiana plants, Robert Benezra (Sloan Kettering Institute) for the gift of plasmid H2-Venus, Max Griesmann (LMU Munich) for providing a new transcriptome assembly for completing the ORFs of DgDef3 and DgDEF4, and Doreen Meier (Philipps-Universität Marburg) for handling the RNA samples for sequencing. Confocal microscopy was performed at the Image Facility of Stockholm University (IFSU). This project was financed by a grant from the Swedish research council Vetenskapsrådet (VR 2012-03061) to KP. The authors acknowledge technical assistance and access to resources supported by BMBF grant FKZ 031A533 within the de.NBI network.

KP, VR 2012-03061, Vetenskapsrådet, vr.se

AB, access to the resources of BMBF grant FKZ 031A533 (network grant for de.NBI); www.bmbf.de

Additional Editor Comments:

Dear Authors,

As suggested by the reviewers, manuscript need a major revision with more clarity in the writing at materials and methods, discussion, and conclusion including abstract and title too.

Please see the detailed comments from the reviewers.

Reviewers' comments:

Reviewer's Responses to Questions

**Comments to the Author**

1. Is the manuscript technically sound, and do the data support the conclusions?

Reviewer #1: Yes

Reviewer #2: Yes

2. Has the statistical analysis been performed appropriately and rigorously? 

Reviewer #1: Yes

Reviewer #2: Yes

3. Have the authors made all data underlying the findings in their manuscript fully available?

Reviewer #1: No

Reviewer #2: Yes

4. Is the manuscript presented in an intelligible fashion and written in standard English?

Reviewer #1: Yes

Reviewer #2: Yes

5. Review Comments to the Author

Reviewer #1: The manuscript by Salgado et al reports the evolutionary relationship and funcational characterization of nodule-specific defensin DgDef1 from an actinorhizal plant Datisca glomerata. Overall the results are quite interesting, but the manuscript is over descriptive, particularly the Materials and Methods sections and require major revision. Please find below my specific comments:

Title need improvement, it can be shorten to precisely emphasizing the main conclusions or posing a question.

Line 50: Promoter GUS studied on transgenic...

Line 56-59: This part need improvment, authors should bring the key conclusions here, rather than descrition of results.

Line 120:Please be specific about the aim of the study, "In this study, we analysed the previously.." analysed for what and why?

Line 163-171: Confusion in this part, authors have studied the gene expression only by RT-qPCR or the abundance of genes was estimated by RNAseq first and then RT-qPCR was performed.

Line 245: crushed nodules of of D. glomerata? please clarify.

Line 247: remove bacteria from ..with Frankia bacteria.

Line 252: Authors cited personal communication on multiple occasions, which is not appropriate. What are these communications, just a preliminary observation not validated or published yet? I believe there might be several reports on similar observations, please cite a peer-reviewed published article.

Whole Materials and Methods section need rewrting, it should be brief and clear, complete protocols can be provided as supllementary information.

Line 462: accession number is missing.

Line 467: Please cite the reference for Demina and collaborators work?

There is no clear distinction between Results and Discussions sections, discussions section seems repetition of Results section.

Rewirte these sections.

Line 612: might suggest?? it appears over speculative expression, and authors should avoid such expressions.

The arbitrary use of NCRs, ANDs and nodule-specific defensins is confusing, authors should revisit the manuscript and make changes to maintain consistency and to avoid confusions.

non-cooperators seems more appropriate term to cheaters.

Line 622: there is no need to define the 'cheaters' again, it was already done in the introduction section.

Line 626: Please substantiate your statement with a few references.

Fig.1 Low bootstrap values were found on majority of the nodes, scale bar is not clear, which model was applied to draw phylogeny.

Fig 3. Why the scale bars were not kept same for all figures?

Reviewer #2: The authors have studied the effect of nodule-specific defensin DgDef1 using a soil actinobacterium, Streptomyces coelicolor A3(2) M145, and two Gram-negative strains, E. coli K-12 substrain MG1655 and the legume microsymbiont Sinorhizobium meliloti 1021. The manuscript is well planned and written. The following comments can be considered before accepting the article.

Add a concluding line highlighting the importance of work in the end of the Abstract.

Even in the last paragraph of the importance and novelty of the present study should be highlighted.

Line 367 – What author mean by commercial production? I think it can be changed.

I suggest all the units of centrifugation can be given in g rather in rpm

Media names when used for the first time should be given in full form.

In IC50, 50 should be subscript.

While mentioning bacterial scientific names full name can be mentioned (Eg: Pichia pastoris). There after it can be mentioned as P. pastoris

6. PLOS authors have the option to publish the peer review history of their article (what does this mean?). If published, this will include your full peer review and any attached files.

Reviewer #1: **Yes: **Praveen Rahi

Reviewer #2: No

---

## [Author Response · Author response to Decision Letter 0]

29 Mar 2022

Responses to reviewers’ comments

Reviewer #1: The manuscript by Salgado et al reports the evolutionary relationship and funcational characterization of nodule-specific defensin DgDef1 from an actinorhizal plant Datisca glomerata. Overall the results are quite interesting, but the manuscript is over descriptive, particularly the Materials and Methods sections and require major revision. Please find below my specific comments:

Title need improvement, it can be shorten to precisely emphasizing the main conclusions or posing a question.

Response: We have changed the title to “Legume NCRs and nodule-specific defensins of actinorhizal plants – do they have a common origin?’

Reviewer #1: Line 50: Promoter GUS studied on transgenic...

Response: This was changed to “Promoter-GUS studies on transgenic hairy roots…”

Reviewer #1: Line 56-59: This part need improvment, authors should bring the key conclusions here, rather than descrition of results.

Response: We have added this sentence: “In combination with the phylogeny results, this suggests that nodule-specific defensin-like peptides were part of the original root nodule symbioses and subsequently lost in most symbiotic legumes, while being maintained in the actinorhizal lineages.”

Reviewer #1: Line 120:Please be specific about the aim of the study, "In this study, we analysed the previously.." analysed for what and why?

Response: This was changed to ”In this study, we analysed the expression pattern and function of the previously…”

Reviewer #1: Line 163-171: Confusion in this part, authors have studied the gene expression only by RT-qPCR or the abundance of genes was estimated by RNAseq first and then RT-qPCR was performed.

Response: We apologize for the confusion. We have solved the problem by introducing a paragraph break between the description of RT-qPCR on plant organs and RT-qPCR to confirm RNAseq results from bacterial cultures, and have added an introductory phrase for the second paragraph: “For validation of RNAseq results on S. meliloti cultures, RT-qPCR measurements were performed in a qTOWER3 G,,,”

Reviewer #1: Line 245: crushed nodules of of D. glomerata? please clarify.

Response: The inoculum was first described in line 139/140; we feel that we should not repeat the explanation. We have, however, extended the original explanation; it reads now: “…and were inoculated with a suspension of nodules (ca. 1 g nodules/L soil). The suspension was prepared from nodules of older D. glomerata plants ground in deioinized water with mortar and pestle (“crushed nodules”).”

Reviewer #1: Line 247: remove bacteria from ..with Frankia bacteria.

Response: Done.

Reviewer #1: Line 252: Authors cited personal communication on multiple occasions, which is not appropriate. What are these communications, just a preliminary observation not validated or published yet? I believe there might be several reports on similar observations, please cite a peer-reviewed published article.

Response: We cited personal communication on two occasions: 

1. Sergio Svistoonoff for the use of 1 mM KNO3 as non-inhibitory nitrate concentration. This reference was deleted.

2. Katharina Markmann for additions to her published protocol. These were additions to the protocol in the peer-reviewed article which we also cite. We have deleted the reference to Katharina Markmann, personal communication, and now only refer to Markmann et al. [44].

Reviewer #1: Whole Materials and Methods section need rewrting, it should be brief and clear, complete protocols can be provided as supllementary information.

Response: We think that a comprehensive description of the methods is relevant for understanding of the results, and do not think it would be at all helpful to distribute the methodological information over two places.

Reviewer #1: Line 462: accession number is missing.

Response: Unfortunately, the accession number was not yet available at the time of submission. It has now been added: “sequences are available at https://www.ebi.ac.uk/arrayexpress/ (accession number E-MTAB-11181).”

Reviewer #1: Line 467: Please cite the reference for Demina and collaborators work?

Response: The reference [45] is cited.

Reviewer #1: There is no clear distinction between Results and Discussions sections, discussions section seems repetition of Results section.

Rewirte these sections.

Response: The results are discussed in the Discussion section. Therefore, some references to the Results section are unavoidable. We have deleted two lines from the first paragraph of the Discussion section which were not essential (“and within this subclade, the four defensins of Datisca glomerata all of which in contrast with other ANDs contain an acidic C-terminal domain, form a distinct group (Figure 1)”.

Reviewer #1: Line 612: might suggest?? it appears over speculative expression, and authors should avoid such expressions.

Response: The complete sentence reads “This finding might suggest many cases of convergent evolution; however, a common evolutionary origin of the nodule-specific cysteine-rich peptides in legumes and actinorhizal plants would seem more parsimonious.” We deliberately use the over-speculative expression in order to make clear that _many_ cases of convergent evolution are _not_ a parsimonious explanation for the facts observed.

Reviewer #1: The arbitrary use of NCRs, ANDs and nodule-specific defensins is confusing, authors should revisit the manuscript and make changes to maintain consistency and to avoid confusions.

Response: 

The use of these terms is not arbitrary. The point which might have given rise to confusion – for which we apologize – is that we sometimes wrote about nodule-specific cysteine-rich peptides (or nodule-specific defensin-like peptides) and sometimes about nodule-specific cysteine-rich proteins. We have changed everything to peptides.

There are two places in this study were both types of nodule-specific cysteine-rich peptides – NCRs and ANDs – are mentioned together: the first chapter of the Results section, and 

the Discussion.

The term “NCRs” has long been defined for the nodule-specific cysteine-rich peptides found in some legumes. The nodule-specific defensins of actinorhizal plants had been, so far, called “nodule-specific defensins”. It is important to note that NCRs do _not_ have the typical cysteine patterns of defensins (there are at least two cysteine residues missing, see Graham et al. 2011 https://doi.org/10.1104/pp.104.037531 and Zorin et al. 2019 https://journals.eco-vector.com/ecolgenet/article/view/11493), however, they are part of the different classes of defensin-like peptides. We define “ANDs” (actinorhizal nodule-specific defensins) in the first sentence of the discussion as the term for the NCR equivalent in actinorhizal plants. 

In the first chapter of results, we list the data that lead to the conclusion that ANDs are, indeed, a common phenomenon in actinorhizal nodules. Thus, the earliest possible point to define ANDs in this manuscript would have been at the end of this chapter. However, we do not think to define a novel acronym in the beginning of the Results section, only to not use it again before the Discussion section, would be a good choice.

The actinorhiza-specific acronym helps to streamline the discussion. The alternative would have been the use of “legume NCRs” and “actinorhizal NCRs”, and given that NCRs do not have the typical cysteine patterns of defensins, while ANDs do, we prefer to use different acronyms for both groups of proteins. When we are talking about both, we use “nodule-specific cysteine-rich peptides”.

We feel that this nomenclature is best suited to avoid misunderstandings.

Reviewer #1: non-cooperators seems more appropriate term to cheaters.

Response: “Cheaters” is the term used in the literature.

Marco et al. 2008 Nature https://www.nature.com/articles/npre.2008.1964.1

Sachs et al. 2010 Journal of Evolutionary Biology https://onlinelibrary.wiley.com/doi/full/10.1111/j.1420-9101.2010.02056.x

Moyano et al. 2017 Math BioSci https://www.sciencedirect.com/science/article/pii/S0025556416303893?via%3Dihub

Walker et al. 2020 Front Plant Sci https://www.frontiersin.org/articles/10.3389/fmicb.2020.585749/full

Reviewer #1: Line 622: there is no need to define the 'cheaters' again, it was already done in the introduction section

Response: Sorry, no. The term “cheater” is not mentioned in the introduction section. Thus, it has to be defined in the discussion.

Reviewer #1: Line 626: Please substantiate your statement with a few references.

Response: This statement is a conclusion based on the distribution of NCR genes in legumes (reference [57]) and otherwise on data presented in this manuscript.

Reviewer #1: Fig.1 Low bootstrap values were found on majority of the nodes, scale bar is not clear, which model was applied to draw phylogeny.

Response: 

We have clarified the scale bars.

We have updated the Methods section, it reads now:

The phylogenetic tree was estimated using RAxML v.8.2.12 [40] using the PROTGAMMAAUTO model and rapid bootstopping (autoMRE) [45].

The cysteine-rich domain sequences are short (ca. 50 amino acids) and does hence not contain much phylogenetic information. Many splits in the phylogeny are short and with low bootstrap support values. We still believe the tree supports our conclusions, as they rely on a relatively well-supported clade (80% bootstrap support). The paragraph in Results now reads: The phylogeny of these cysteine-rich domains was inferred using defensins and cysteine-rich peptides from non-symbiotic plant species as well as nodule-specific cysteine-rich peptides (NCRs) from legumes and nodule-specific defensins from actinorhizal plants [26,27,29]. The resulting tree (Figure 1) showed that the NCRs of legumes and the defensins of actinorhizal species are part of the same relatively well-supported clade (80% bootstrap, see star in Figure 1). This clade included nodule-specific defensins from all orders of symbiotic plants: Fabales (legumes [26]), Fagales (Alnus glutinosa and Casuarina glauca [27]), Rosales (Ceanothus thyrsiflorus [28]) and Cucurbitales (D. glomerata) (Figure 1). 

Reviewer #1: Fig 3. Why the scale bars were not kept same for all figures?

Response: The scale bar is the same for all figures.

Reviewer #2: The authors have studied the effect of nodule-specific defensin DgDef1 using a soil actinobacterium, Streptomyces coelicolor A3(2) M145, and two Gram-negative strains, E. coli K-12 substrain MG1655 and the legume microsymbiont Sinorhizobium meliloti 1021. The manuscript is well planned and written. The following comments can be considered before accepting the article.

Add a concluding line highlighting the importance of work in the end of the Abstract.

Response: We have concluded the abstract with “Overall, the changes induced by DgDef1 are reminiscent of those of some legume NCRs, suggesting that nodule-specific defensin-like peptides were part of the original root nodule toolkit and were subsequently lost in most symbiotic legumes, while being maintained in the actinorhizal lineages.”

Reviewer #2: Even in the last paragraph of the importance and novelty of the present study should be highlighted.

Response: The first sentence of the last paragraph of the introduction reads now: “To understand whether legume NCRs and nodule-specific defensins from actinorhizal nodules represent an example of convergent evolution or share a common origin, we analysed the phylogeny, expression pattern and function of the previously reported nodule-specific defensin DgDef1 from the actinorhizal plant D. glomerata (Cucurbitaceae, Cucurbitales) [28].” 

Reviewer #2: Line 367 – What author mean by commercial production? I think it can be changed.

Response: The headline of this chapter has been changed to “Synthetic DgDef1 without the CTPP domain (DgDef1𝛥SP𝛥CTPP)”. The first sentence of the chapter reads: “DgDef1𝛥SP𝛥CTPP was chemically synthesized by conventional solid phase peptide synthesis yielding a purity of 95.54% reported by HPLC analysis (ProteoGenix SAS, France).” We hope this is clear now.

Reviewer #2: I suggest all the units of centrifugation can be given in g rather in rpm

Response: Done.

Reviewer #2: Media names when used for the first time should be given in full form.

Response: Corrected.

Reviewer #2: In IC50, 50 should be subscript.

Response: Corrected.

Reviewer #2: While mentioning bacterial scientific names full name can be mentioned (Eg: Pichia pastoris). There after it can be mentioned as P. Pastoris

Response:

The model yeast Pichia pastoris is first mentioned (genus name spelled out) in line 142 (MatMet). Afterwards, the genus name is spelled out in headlines and figure legends, but not in the text (we have corrected that in line 317). The genus name is spelled out again in line 531, i.e., at first mention in the Results section.

---

## [Decision Letter · Decision Letter 1]

6 May 2022

Legume NCRs and nodule-specific defensins of actinorhizal plants – do they share a common origin?

PONE-D-21-31806R1

Dear Dr. Pawlowski,

We’re pleased to inform you that your manuscript has been judged scientifically suitable for publication and will be formally accepted for publication once it meets all outstanding technical requirements.

Kind regards,

Dharam Singh

Academic Editor

PLOS ONE

Additional Editor Comments (optional):

All concerns have been satisfactorily addressed. The revised version of the manuscript has been improved dramatically.

Reviewers' comments:

Reviewer's Responses to Questions

**Comments to the Author**

1. If the authors have adequately addressed your comments raised in a previous round of review and you feel that this manuscript is now acceptable for publication, you may indicate that here to bypass the “Comments to the Author” section, enter your conflict of interest statement in the “Confidential to Editor” section, and submit your "Accept" recommendation.

Reviewer #1: All comments have been addressed

Reviewer #2: All comments have been addressed

2. Is the manuscript technically sound, and do the data support the conclusions?

Reviewer #1: Yes

Reviewer #2: Yes

3. Has the statistical analysis been performed appropriately and rigorously? 

Reviewer #1: Yes

Reviewer #2: Yes

4. Have the authors made all data underlying the findings in their manuscript fully available?

Reviewer #1: Yes

Reviewer #2: Yes

5. Is the manuscript presented in an intelligible fashion and written in standard English?

Reviewer #1: Yes

Reviewer #2: Yes

6. Review Comments to the Author

Reviewer #1: I can see that authors have addressed all the queries raised, including modifying the title of the manuscript.

Reviewer #2: This article was greatly improved compared with the previous manuscript. I believe it has met the request of the Journal, and could be accepted.

7. PLOS authors have the option to publish the peer review history of their article (what does this mean?). If published, this will include your full peer review and any attached files.

Reviewer #1: **Yes: **Praveen Rahi

Reviewer #2: No

---

## [Editor Report · Acceptance letter]

4 Aug 2022

PONE-D-21-31806R1 

Legume NCRs and nodule-specific defensins of actinorhizal plants – do they share a common origin? 

Dear Dr. Pawlowski:

I'm pleased to inform you that your manuscript has been deemed suitable for publication in PLOS ONE. Congratulations! Your manuscript is now with our production department. 

Kind regards, 

on behalf of

Dr. Dharam Singh 

Academic Editor

PLOS ONE